# GPS Pipeline: portable, scalable genomic pipeline for *Streptococcus pneumoniae* surveillance from Global Pneumococcal Sequencing Project

Harry C. H. Hung [1] ✉, Narender Kumar[1], Victoria Dyster[1], Corin Yeats [2], Benjamin Metcalf [3], Yuan Li [3], Paulina A. Hawkins [3], Lesley McGee [3], Stephen D. Bentley [1] & Stephanie W. Lo [1,4,5] ✉

*Streptococcus pneumoniae* (pneumococcus) is a major pathogen globally, responsible for an estimated one million deaths annually and contributing significantly to the global burden of antimicrobial resistance. Ongoing surveillance of its vaccine antigen (i.e. serotypes), antimicrobial resistance, and pneumococcal lineages is crucial for assessing the impact of vaccination programs and guiding future vaccine design. However, current bioinformatics tools have several limitations that prevent them from enabling comprehensive analysis that allows simultaneous, large-scale, and independent generation of these crucial data. Here, we present the GPS Pipeline that enables reliable extraction of public health information from pneumococcal genomes using in silico methods. It can accurately identify 102 of 107 known serotypes, recognise 1053 pneumococcal lineages, and predict susceptibilities to 19 common antibiotics. Built on Nextflow and utilising containerisation technology, the GPS Pipeline minimises software setup requirements and bioinformatics expertise while facilitating large-scale analysis of genomic data. The GPS Pipeline was applied and validated on 20,924 pneumococcal genomes worldwide, demonstrating its effectiveness in enhancing responsiveness in pneumococcal genomic surveillance.

*Streptococcus pneumoniae* (pneumococcus) is a globally significant pathogen and the predominant cause of lower respiratory infections, responsible for over one million deaths worldwide each year[1]. The introduction of pneumococcal vaccines has substantially reduced the incidence of pneumococcal disease and mitigated antimicrobial resistance (AMR) associated with antibiotic treatments[2]. However, vaccines do not target all serotypes, and serotype switching facilitates

pneumococci to evade vaccines while acquiring additional AMR[3]. Thus, leveraging genomics for pneumococcal disease surveillance and guiding vaccine strategies, as demonstrated by the Global Pneumococcal Sequencing (GPS) project, is crucial[4]. The increasing accessibility of whole-genome sequencing (WGS) has led to a rapid expansion of publicly available pneumococcal genomes (Fig. 1), providing an unprecedented opportunity for comprehensive disease surveillance

[1]Parasites and Microbes, Wellcome Sanger Institute, Hinxton, UK. [2]Centre for Genomic Pathogen Surveillance, Pandemic Sciences Institute, University of Oxford, Oxford, UK. [3]Division of Bacterial Diseases, Centers for Disease Control and Prevention, Atlanta, USA. [4]Milner Centre for Evolution, Department of Life Sciences, University of Bath, Bath, UK. [5]The Great Ormond Street Institute of Child Health, University College London, London, UK. ✉e-mail: ch31@sanger.ac.uk; sl28@sanger.ac.uk

**Fig. 1 | The number of *Streptococcus pneumoniae* samples available on the European Nucleotide Archive (ENA) has been steadily rising over the years.** Metadata of samples with NCBI Taxonomy ID of *Streptococcus pneumoniae* (1313) were retrieved from the ENA Browser on 9th April 2024. Annual published genomes from Year 2010 to 2023 were computed based on the "first_public" field in the metadata. While the number of genomes does not consistently increase every year, there is an overall upward trend of publication of pneumococcal genomes.

through integrative analysis to infer serotypes, antimicrobial resistance and lineages simultaneously.

Conducting genomic surveillance of pathogens requires a unique cross-disciplinary blend of knowledge in epidemiology, clinical microbiology, bioinformatics and IT infrastructure, in order to analyse the genomic data at scale and to understand the implication of the analysis results. However, there often exists a gap in bioinformatics expertise, especially in low- and middle-income countries (LMICs)[5]. At the same time, LMICs are the most affected by disease due to pneumococcus, one of the leading causes of death in children under 5 years old[6].

While numerous projects, including the GPS project, have set out to assist the development of bioinformatics expertise and infrastructure in LMICs in the long term, it will take time for these efforts to come to fruition. One possible alternative approach to address these gaps could be the development of easy-to-use bioinformatics pipelines and tools. This approach would also benefit regions with more advanced bioinformatics capabilities by further enhancing their efficiency in processing genomic data. While well-known pipelines and tools exist, their constraints, such as difficulties in deployment and insufficient pneumococcus-specific analysis capabilities, may hinder effective use for the pneumococcal surveillance in LMICs.

One of the first pneumococcal-specific pipelines is the U.S. Centres for Disease Control and Prevention (CDC) typing pipeline developed by the *Streptococcus* Laboratory[7]. It is an automated Perl-based pipeline that performs whole-genome sequencing-based strain characterisation, including deduction of capsular serotype, pilus type, multilocus sequence type (MLST), penicillin-binding protein (PBP) sequence type, and minimum inhibitory concentration (MIC) predictions for 18 antibiotics. This typing pipeline is primarily used for CDC's domestic US-based Active Bacterial Core Surveillance (ABCs) programme[8] and several research groups outside the CDC. Strain characterisations generated by the CDC pipeline were comprehensively validated using phenotypic testing results from ABCs isolates[7,9,10]. The implementation of the CDC pipeline is an in-house script without detailed installation instructions and is not containerised; thus, its installation often requires customisation to local

computational environments and additional bioinformatics expertise. Efforts to modernise this pipeline by refactoring it into a Nextflow pipeline, which promises to improve the ease of deployment, are still under development[11].

A second pipeline, Bactopia, is an all-encompassing bacterial genomic Nextflow pipeline with additional workflows called Bactopia Tools that can carry out a wide range of analysis with lots of flexibility[12]. However, due to its flexibility, it requires additional configuration and customisation by the user to generate relevant information for pneumococcal surveillance. This characteristic makes it unsuitable for quick deployment. Another limitation of Bactopia is its lack of integration with PopPUNK[13], which is a tool widely used in the pneumococcal research community for lineage assignment (i.e., Global Pneumococcal Sequence Cluster, GPSC)[14].

Pathogenwatch[15], a web application designed for genomic surveillance, allows researchers to upload reads or genome assemblies of samples along with metadata to its website, where processing is performed on its servers. The results are then accessible and downloadable on the website. While it is a powerful and easy-to-use tool for genomic surveillance requiring minimal infrastructure, it requires a stable internet connection, which may not be available in some LMICs. As the data is processed remotely, researchers need to upload a large amount of data for each sample, which might not be feasible in areas that have slow and unreliable internet service. While Pathogenwatch provides detailed technical descriptions and open-sourced components, it is not designed to be deployed by users without web infrastructure expertise. It limits the user's ability to run their own instance locally.

PneusPage[16] is another web application for pneumococcal genome analysis that enables researchers to upload reads for automated analysis on its server via a web interface. Analysis results are provided on the website, including quality control metrics and a range of in silico typing outputs. Like Pathogenwatch, PneusPage has minimal infrastructure requirements but depends heavily on a stable internet connection, which limits its usability in settings with limited internet access. In addition, the application has only been validated on a relatively small dataset of 80 genomes, and its server is designed to

**Table 1 | Features of GPS Pipeline, and other similar bioinformatics pipelines and tools**

| Features | GPS Pipeline | CDC pipeline[7] | Bactopia[12] | Pathogenwatch[15] | PneusPage[16] |
|---|---|---|---|---|---|
| Ready-to-use | Yes, only one-off initialisation required | No, designed for in-house use | No, requires additional configuration | Yes, no setup required | Yes, no setup required |
| Require internet | Only during initialisation | No | Only during initialisation | Yes | Yes |
| Local processing | Yes | Yes | Yes | No | No |
| Open source | Yes | Yes | Yes | Yes | No |
| Validated with pneumococcal genomes | Yes, validated with 20,924 genomes | Yes, validated with 2316 genomes | Not stated | Not stated | Yes, validated with 80 genomes |
| Accepted input type | Illumina reads | Illumina reads | Assemblies, Illumina and Nanopore reads | Assemblies and Illumina reads | Illumina reads |
| Quality control check | Yes | No | Yes | Yes | Yes |
| Serotype prediction | Yes | Yes | Yes | Yes | Yes |
| MLST prediction | Yes | Yes | Yes | Yes | Yes |
| Lineage assignment | Yes, GPSC | No | No | Yes, GPSC | Yes, GPSC |
| AMR prediction | Yes, with resistance category prediction | Yes, with resistance category prediction | Partial, only perform AMR genes detection | Yes, with resistance category prediction | Yes, with resistance category prediction |
| Virulence prediction | Yes | No | Yes | No | Yes |

process a maximum of 10 concurrent jobs. These limitations raise concerns about its scalability for large-scale genomic surveillance. Furthermore, as the codebase is not open source, researchers cannot deploy local instances. This poses a significant limitation in terms of reproducibility and data sovereignty.

In this work, we built an intuitive, portable, all-in-one pipeline to analyse pneumococcal genomes, generating key public-health information, including in silico serotypes, pneumococcal lineages (i.e., GPSC), MLST, and antimicrobial susceptibilities against 19 commonly used antibiotics, all at scale with a reasonable turnaround time to overcome the current challenges (Table 1).

## Results

### Demonstration of the portability of the pipeline

The GPS Pipeline itself is < 5 MB in size. The default databases, which are downloaded and uncompressed on the first run, use about 19GB of disk space. The container images disk size is ~13 GB with Docker or ~4.5 GB with Singularity. Hence, the essential files to run the pipeline occupy only 23.5 − 32 GB of space in total. To benchmark the versatility and performance on various platforms, we processed 100 QC-passed genomes in the GPS database on a 16-core Ubuntu-based OpenStack instance running on Intel Xeon Gold 6226 R processors, and 500 genomes (400 QC-passed and 100-QC failed) on the LSF-based Wellcome Sanger Institute Farm5 high-performance computing (HPC) clusters running on a mixture of Intel Xeon Gold and AMD EPYC processors. It took 2 h 48 min and 1 h 40 min (depending on cluster availability), respectively.

### Validation of the pipeline outputs against GPS database in silico typing results

We validated robustness and accuracy of the pipeline by running it on all published samples of the GPS database (*n* = 20,924, last accessed on 14th June 2024), all of which passed quality control, were in silico-typed, and verified to have high co-concordance with phenotypic data[14]. In the GPS database, the genomes were previously assembled by a Perl pipeline written by the Wellcome Sanger Institute Pathogen Informatics Team[17], composed of Velvet version 1.2.10 [18], VelvetOptimiser version 2.2.5[19], SSPACE version 2.0[20], GapFiller version v1.11[21] and SMALT version 0.7.4[22]. For in silico typing, mlst_check version 2.1.1630910 [23] was used to assigned MLST, PneumoCaT v1.2[24] or SeroBA v1.0.0[25] was used to infer serotypes for data generated before and after 15th June 2018, respectively. The CDC pneumococcal typing pipeline (which includes CDC PBP AMR Predictor) was used to detect

AMR with supplementary screening by ARIBA version 2.14.4[26], and GPSC was assigned with PopPUNK version 1.1.5 with GPSC reference database v4.

All genomes from the published GPS database were successfully processed by the pipeline, with 98.0% (20,506/20,924) of the genomes considered to be QC passed. Although all samples passed the read QC, some failed in the assembly, mapping, or taxonomy QC, or a combination of these. The failures are primarily due to tightened QC parameters (Fig. 2) that exclude low-quality genomes to prevent negatively impacting downstream analyses. Most of the genomes failed by the GPS Pipeline showed non-pneumococcal reads contamination (non-*Streptococcus* genus > 2%), multiple pneumococcal strains in a single sample (heterozygous single nucleotide polymorphism (het-SNP) sites > 220). These factors lead to an inflated assembly length exceeding 2.3 Mb.

The 20,506 QC-passed genomes were subject to in silico typing including 14 results shared by the GPS database and the GPS Pipeline: (1) GPSC; (2) MLST; (3) serotype; (4) PBP type and its inferred resistance category of 6 β-lactam antibiotics (amoxicillin (AMO), ceftriaxone (CFT), cefotaxime (TAX), cefuroxime (CFX), meropenem (MER), and penicillin (PEN)); (5-14) 10 individual predicted resistance category of 9 non-β-lactam antibiotics (chloramphenicol (CHL), clindamycin (CLI), co-trimoxazole (COT), doxycycline (DOX), erythromycin (ERY), fluoroquinolones (FQ), levofloxacin (LFX), rifampin (RIF), and tetracycline (TET)), and of both ERY and CLI simultaneously. Predictions of 4 additional antibiotics (kanamycin (KAN), sulfamethoxazole (SMX), trimethoprim (TMP), and vancomycin (VAN)) are available in the GPS Pipeline but not in the GPS database, and therefore not included in the subsequent comparison. Among all genomes typed by the GPS Pipeline, 91.8% (*n* = 18,819/20,506) have the identical results in the GPS database for all shared in silico typing results. For the remaining 1687 genomes, they shared 1898 changes in total (Table 2).

For the GPSC assignment, 0.72% (*n* = 147/20,506) of QC-passed genomes received a different result in the GPS Pipeline when compared to the GPS database. They are either an assignment of previously unassigned genomes or an assignment update due to a cluster merging event, as the latest PopPUNK GPS database used by the pipeline covers more lineages.

For the MLST assignment, the changes impacted only 0.21% (*n* = 43/20,506) of genomes. They were a mixture of failure or change in assignments, and assignment of previously unassigned genomes. It is likely caused by the switch from mlst_check to mlst[27] as the tool for MLST assignment in the pipeline, which has a different assignment algorithm.

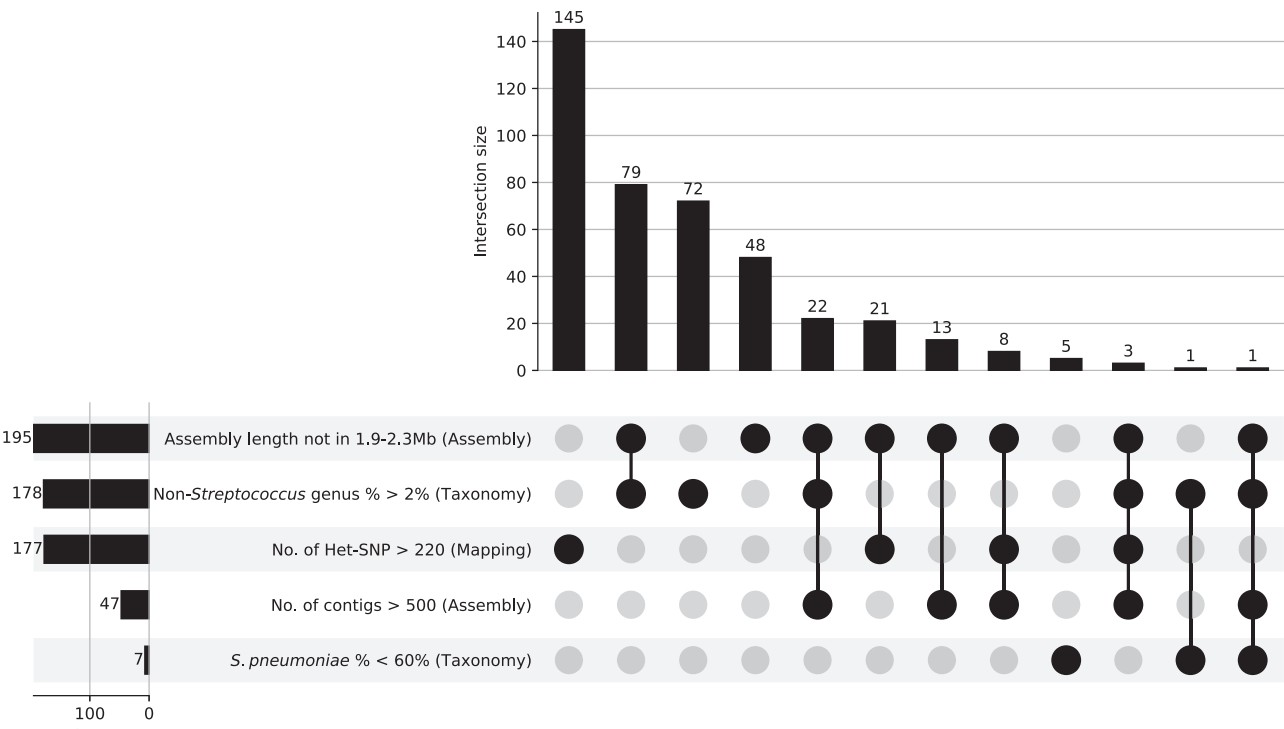

**Fig. 2 | Reasons for 418 pneumococcal genomes failing QC by the GPS Pipeline.** The UpSet plot shows the number of samples involved in each QC parameter failure in rows, and the number of samples in each combination of QC parameter failure in columns. A total of 418 samples failed in 604 QC parameters. Among failed samples, 84.2% ($n = 352/418$) failed in at least one of the new or updated QC parameters: non-*Streptococcus* genus percentage, and number of non-cluster heterozygous SNP (Het-SNP), suggesting the majority of failures are caused by tightened QC requirements.

For serotype prediction, 2.37% ($n = 486/20,506$) of genomes were predicted to have a different serotype. Most of them were changes within the same serogroup or the adoption of the latest naming conventions, followed by improvements in detection precision. They could either be the outcome of the transition from the combined use of PneumoCat and SeroBA in the database to exclusive reliance on SeroBA v2.0.4[28] in the pipeline. SeroBA v2.0.4, the latest serotyping method developed by our team, is capable of recognising 102 of 107 known pneumococcal serotypes, as well as 4 groups of non-encapsulated pneumococci.

For PBP type assignment and β-lactam antibiotics resistance category prediction, 0.36% ($n = 73/20,506$) of genomes observe various changes. The majority of the changes involved alterations in PBP type, most of which led to the failure to assign a previously assigned resistance category ($n = 30/20,506$) or resulted in the assignment of a previously unassigned resistance category ($n = 20/20,506$). It is potentially due to the change of de novo assembler between the database and the pipeline, from a custom Velvet-based Perl pipeline to the SPAdes-based Shovill[29,30]. As assemblies instead of sequence read files are passed to the CDC PBP AMR Predictor for typing and prediction, where changes in assemblies resulting from the shift from a Velvet-based assembler to a Shovill-based one which could generate higher quality assemblies (Table 3) may change the prediction results.

For the non-β-lactam antibiotics resistance category prediction, 4.87% ($n = 998/20,506$) of genomes observed 1149 changes in total. The changes are anticipated due to the move from the CDC pneumococcal typing pipeline in the database to a custom ARIBA-based prediction in the pipeline. As we have reimplemented the detection mechanism, updated the target genomic changes and mobile elements based on the latest publications, and adjusted detection thresholds to avoid false positives.

## Validation of the pipeline outputs against GPS database experimental phenotype results

To further assess the accuracy of the GPS Pipeline, we compared its in silico predictions to experimental phenotypic results recorded in the GPS database, when available (Table 4). These comparisons were carried out for serotype, and for resistance category to β-lactam and non-β-lactam antibiotics.

For serotype prediction, 11,810 of the 20,506 QC-passed genomes had corresponding phenotypic serotype records determined by the Quellung reaction. Since phenotypic data may report serogroups or multiple possible serotypes, we defined concordance as: (1) an exact serotype match; (2) the in silico prediction falling within the reported phenotypic serogroup; or (3) the prediction matching one of the multiple serotypes listed. Using these criteria, 89.32% ($n = 10,549/11,810$) of genomes showed concordance between the GPS Pipeline output and phenotypic data.

For β-lactam resistance, 1273 genomes had MIC results, measured by the broth dilution method, available for all 6 β-lactam antibiotics assessed by the pipeline (AMO, CFT, TAX, CFX, MER, PEN). Phenotypic test results were interpreted into resistance categories according to the 2014 Clinical & Laboratory Standards Institute (CLSI) guideline interpretation[31], and compared to GPS Pipeline predictions. The average concordance across all 6 antibiotics was 94.36%, with the lowest individual concordance being penicillin at 88.22% ($n = 1123/1273$).

For non-β-lactam resistance, 8 antibiotics (CHL, CLI, COT, ERY, LFX, RIF, TET, VAN) are shared between the GPS Pipeline output and the phenotypic test data available in the GPS database. A total of 3117 genomes had MIC results, measured by the broth dilution method, for all 8 antibiotics. Using the 2014 CLSI guideline interpretation, the average concordance between GPS Pipeline predictions and resistance categories of phenotypic results was 91.06%. RIF and TET showed notably lower concordance rates of 77.19% ($n = 2406/3117$) and 64.97%

**Table 2 | Statistics of changes in 14 in silico typing categories in the GPS Pipeline when compared to the GPS Database**

| In silico Typing Categories | Description of Change | Count |
|---|---|---|
| GPSC [a] (n = 147) | Assignment of previously unassigned genomes | 146 |
| | Updated assignment due to the GPSC cluster merging event | 1 |
| MLST [b] (n = 43) | Failure to assign a previously assigned genome | 22 |
| | Change in assignment | 13 |
| | Assignment of previously unassigned genomes | 8 |
| Serotype (n = 486) | Change in assignment within the same serogroup | 169 |
| | Update of serotype name (e.g., 10X → 33 G, Swiss_NT → NCC2_aliC_aliD_non_encapsulated) | 141 |
| | Assignment of previously untypable nonencapsulated (e.g., NCC1_pspK_non_encapsulated) | 86 |
| | Improved assignment for subtype (e.g., 20 → 20 A/20B, 36 → 36 A/36B) | 41 |
| | Failure to assign a previously assigned genome | 34 |
| | Change in assignment to a different serogroup | 15 |
| PBP [c] type and its resulting resistance category of 6 β-lactam antibiotics (n = 73) | Change in PBP type; failure to assign previously assigned resistance category | 30 |
| | Change in PBP type; assignment of previously unassigned resistance category | 20 |
| | No change in PBP type; change in resistance category | 11 |
| | Change in PBP type; change in resistance category | 7 |
| | Change in PBP type; no change in resistance category | 5 |
| CHL [d] resistance category (n = 17) | Change from susceptible to resistant | 13 |
| | Change from resistant to susceptible | 4 |
| CLI [e] resistance category (n = 24) | Change from resistant to susceptible | 22 |
| | Change from susceptible to resistant | 2 |
| COT [f] resistance category (n = 157) | Change from susceptible to intermediate | 62 |
| | Change to indeterminable due to low gene coverage | 51 |
| | Change from resistant to intermediate | 33 |
| | Change from intermediate to susceptible | 6 |
| | Change from intermediate to resistant | 5 |
| DOX [g] resistance category (n = 37) | Change from resistant to susceptible | 37 |
| ERY [h] resistance category (n = 215) | Change from resistant to susceptible | 190 |
| | Change from susceptible to resistant | 25 |
| ERY & CLI resistance category (n = 23) | Change from resistant to susceptible | 21 |
| | Change from susceptible to resistant | 2 |
| FQ [i] resistance category (n = 189) | Change from intermediate to susceptible | 97 |
| | Change from resistant to susceptible | 54 |
| | Change from susceptible to intermediate | 37 |
| | Change from susceptible to resistant | 1 |
| LFX [j] resistance category (n = 364) | Change from susceptible to intermediate | 293 |
| | Change from susceptible to resistant | 71 |
| RIF [k] resistance category (n = 86) | Change from susceptible to resistant | 86 |
| TET [l] resistance category (n = 37) | Change from resistant to susceptible | 37 |

[a] Global Pneumococcal Sequence Cluster; [b] Multilocus sequence typing; [c] Penicillin-binding protein; [d] Chloramphenicol; [e] Clindamycin; [f] Co-trimoxazole; [g] Doxycycline; [h] Erythromycin; [i] Fluoroquinolones; [j] Levofloxacin; [k] Rifampin; [l] Tetracycline.

(n = 2025/3117), respectively. This discrepancy is primarily due to the presence of the intermediate resistance category in phenotypic results, while no known genes or mutations currently explain intermediate resistance to RIF or TET. As a result, the GPS Pipeline cannot assign the intermediate category for these antibiotics. Intermediate calls accounted for 21.69% (n = 686/3117) of RIF and 33.21% (n = 1035/3117) of TET phenotypic results. When intermediate calls are excluded from the comparison, the concordance for RIF and TET increases to 98.97% (n = 2406/2431) and 97.26% (n = 2025/2082), respectively.

### Example workflow of using the pipeline in a country genomic surveillance analysis

To demonstrate that the GPS Pipeline can easily generate useful in silico typing results for genomic surveillance, we acquired 263 published genomes from Poland, and their associated sample metadata (Supplementary Data 1), from the Monocle Data Viewer (data-viewer.monocle.sanger.ac.uk/project/gps). We processed all genomes with the GPS Pipeline with default parameters on a 16-core Ubuntu-based OpenStack instance, completing in 7 h 23 min without user intervention. Of the 263 genomes, 262 passed quality control and were successfully in silico typed by the pipeline (Supplementary Data 2). The 262 QC-passed genomes were used to generate a phylogenetic tree using snippy v4.6.0[32], SNP-sites v2.5.1[33], and FastTree 2.1.11[34]. We then merged the sample metadata with the pipeline results and uploaded the combined dataset and the phylogenetic tree to Microreact[35] to produce an informative visualisation of lineage, serotype, and AMR distribution and evolution in Poland (Fig. 3). The created Microreact instance is available at microreact.org/project/8RYZQHebzFo3qGkgQ1hdd3. This demonstrated the GPS Pipeline can efficiently generate valuable public health insights from genomic data for surveillance and analysis purposes.

### Real-world adaptations of the pipeline

Since the GPS Pipeline reached feature-complete status and validated, we have used it to process more than 20,000 new genome submissions to the GPS project for quality control and in silico typing. The pipeline's standardised and consistent outputs facilitate straightforward integration of information on new genomes into the GPS project database and enable published genomes to be readily made available on the Monocle Data Viewer. In addition, the pipeline and its generated results are starting to be used in publications, preprints, and presentations[36–38], demonstrating their usefulness to researchers in real-world scenarios.

## Discussion

A portable and adaptable pipeline for analysing *S. pneumoniae* genomes was built to extract key public health information, including in silico serotype, GPSC, MLST and prediction of antimicrobial susceptibility to 19 common antibiotics, with a minimum requirement of IT infrastructure. The design of the pipeline allows users to analyse data ranging from HPC clusters (e.g., the GPS project at Sanger) to a laptop, yet retaining high reproducibility. The standardised in silico typing output from the GPS Pipeline could be easily adapted to be displayed on Microreact - an interactive web application to visualise epidemiological data and phylogeny, and well-suited for subsequent downstream processing and statistical analysis in R or other statistical software. The GPS core team and partners have used this pipeline to process > 20,000 pneumococcal genomes since 2023, and the in silico typing output, combining the epidemiological metadata, are streamlined to upload to the project database, minimising the risk of manual errors. This demonstrates the GPS Pipeline's capability to analyse data at scale and in real-time, highlighting its potential for full automation, from processing raw sequencing data to generating comprehensive reports.

**Table 3 | Performance and quality comparison between de novo assemblers**

| Sequencing System | Assembler | Performance metrics (mean ± 95% CI) | | Assembly quality metrics (mean ± 95% CI) | | |
|---|---|---|---|---|---|---|
| | | Run Time (seconds) | Maximum memory usage (MB) | No. of contigs (≥ 500 bps) | N50 | Assembly length not in 1.9 – 2.3 Mbps |
| Illumina MiSeq (n = 10) | Shovill | 255.47 ± 153.13 | 2997.71 ± 463.42 | 57.60 ± 21.13 | 97883.00 ± 29008.98 | 0 |
| | Unicycler | 689.11 ± 307.04 | 4284.82 ± 1389.25 | 55.10 ± 12.39 | 101738.10 ± 28805.49 | 0 |
| | VelvetOptimiser | 406.49 ± 152.41 | 1271.56 ± 684.86 | 185.50 ± 107.66 | 33612.40 ± 12354.01 | 1 |
| Illumina HiSeq (n = 10) | Shovill | 176.12 ± 53.84 | 3542.72 ± 476.95 | 60.10 ± 17.73 | 76842.80 ± 16158.96 | 0 |
| | Unicycler | 723.20 ± 246.76 | 5566.50 ± 1267.15 | 60.50 ± 13.40 | 77030.80 ± 14694.38 | 0 |
| | VelvetOptimiser | 578.53 ± 176.60 | 6085.60 ± 1996.64 | 90.40 ± 31.81 | 38519.50 ± 17710.91 | 2 |
| Illumina NextSeq (n = 10) | Shovill | 157.25 ± 31.35 | 3764.99 ± 534.04 | 46.10 ± 11.60 | 88366.10 ± 18882.47 | 0 |
| | Unicycler | 804.63 ± 203.17 | 5734.37 ± 1592.26 | 47.00 ± 11.79 | 133739.20 ± 77563.12 | 0 |
| | VelvetOptimiser | 838.78 ± 293.64 | 2315.92 ± 864.58 | 114.40 ± 50.92 | 18549.30 ± 11195.64 | 4 |
| Illumina NovaSeq (n = 10; n = 9 for VelvetOptimiser *) | Shovill | 117.60 ± 23.65 | 4219.44 ± 844.23 | 37.20 ± 7.70 | 173587.00 ± 83611.64 | 0 |
| | Unicycler | 573.42 ± 204.64 | 5022.16 ± 1597.11 | 38.90 ± 9.51 | 177481.30 ± 87313.33 | 0 |
| | VelvetOptimiser | 435.33 ± 184.84 | 3067.10 ± 1437.43 | 136.56 ± 106.57 | 45634.33 ± 12066.71 | 1 |

*VelvetOptimiser repeatedly crashed while assembling one of the samples.

The GPS Pipeline was tested by >15 research groups worldwide; Notifications for problems and overall feedback collected from users over the past year were addressed and taken into consideration in the development of the current version. The pipeline has been well received, with some bug reports in the early phase of development, which were promptly resolved as documented at the issues section in GitHub. We also received reports of difficulties initialising the pipeline in regions with unreliable internet service due to the large size of the databases. To address this challenge, we have worked with the developers of PopPUNK to generate a reduced-size PopPUNK database that brought the total database size from 30GB down to 19GB without a performance penalty. Most of the recently reported problems were related to setting up dependencies required by Nextflow or Docker, particularly Java. We therefore added links to guides on how to set up the necessary dependencies of the pipeline to its manual in the GitHub repository (README.md).

Based on user feedback, once the pipeline is successfully set up, it stably processes the raw read files of samples with ease. This includes users running the pipeline on their laptops, institute workstations, virtual machines on institutional supercomputers, as well as HPC clusters. The GPS team has accomplished the objective of developing a portable and adaptable analysis pipeline for *S. pneumoniae* while remaining user friendly.

A key technical limitation of the GPS Pipeline is its exclusive compatibility with Illumina paired-end short reads, rendering it incompatible with sequencing data from other platforms. This is because several of the bioinformatics tools integrated into the pipeline, particularly SeroBA, which is the most up-to-date tool for pneumococcal serotyping, only support Illumina paired-end short reads. While this restricts the versatility of the pipeline, 99.6% (n = 42,291/42,452) of pneumococcal raw reads publicly released in 2024 were generated using the Illumina platform, according to records from the European Nucleotide Archive (ENA)[39], suggesting that this limitation has limited practical impact. As the pipeline is built on Nextflow and containerisation technology, making it highly modular by design. Therefore, if appropriate tools that support alternative sequencing platforms become available, they can be integrated as an alternative or replacement with relatively short development time. A current workaround, if only in silico typing is needed without accurate quality control, is to first assemble the reads from other sequencing platforms and then convert them into simulated Illumina reads as the input to the pipeline. According to collaborator feedback, this approach can successfully enable in silico typing of long-read samples.

Another limitation of the GPS Pipeline is its inability to predict the intermediate resistance category for RIF and TET. This is because no genes or mutations have yet been identified that explain intermediate resistance to these antibiotics, and similar limitations exist in comparable tools. However, once the underlying mechanisms of intermediate resistance are discovered, they can be rapidly incorporated into the pipeline.

The GPS Pipeline allows scientists with basic computer skills to set it up and analyse independently generated pneumococcal sequence data with zero configuration and without the need for advanced computing resources. It readily extracts public-health relevant bacterial strain information to predict and evaluate the impact of pneumococcal conjugate vaccine on the pneumococcal population, thereby providing important information for the development of vaccine strategies. The automation of the pipeline enhances surveillance responsiveness. Its design, which is modularised and containerised, ensures portability, reproducibility, and simplifies future updates on bioinformatics tools. In addition, it facilitates the expansion of databases for detecting new serotypes and antimicrobial resistance.

## Methods
### Architecture
To ensure portability, reproducibility, and easy deployment, the GPS Pipeline (Fig. 4 for schematic overview; Supplementary Fig. 1 for technical flowchart) was designed and implemented in Nextflow v24.10.3[40] within containerised environments deployed by Docker[41] as default or by Singularity[42] as an alternative option. The pipeline has a single-entry point for inputting the directory containing raw paired-end short read FASTQ files by Illumina paired-end short read sequencing (Illumina, San Diego, CA, USA) of pneumococcal genomes, then the FASTQ files are automatically processed by multiple Nextflow processes in parallel. The pipeline output includes (1) a single CSV file named results.csv containing all quality control (QC) and in silico typing results, including in silico serotypes, GPSC, MLST, and antimicrobial susceptibilities to 19 common antibiotics in MIC value with CLSI guideline interpretation, (2) de novo assemblies in the format of FASTA files, and (3) a text file named info.txt that includes runtime information and software versions.

The pipeline can be run without requiring installation or administrative privileges on any POSIX-compatible system, provided that Bash 3.2 or above, Java 11 or above, and either Docker or Singularity are pre-installed. Compatible operating systems include Linux, Windows

**Table 4 | Statistics of concordance between the GPS Pipeline in silico outputs and GPS database experimental phenotype results**

| Typing Categories | Sample Count | Typing Item | Concordance Count | Concordance Ratio |
|---|---|---|---|---|
| Serotype | 11,810 | Serotype | 10,549 | 89.32% |
| β-lactam resistance category | 1273 | AMO [a] resistance category | 1223 | 96.07% |
| | | CFT [b] resistance category | 1185 | 93.09% |
| | | CFX [c] resistance category | 1236 | 97.09% |
| | | MER [d] resistance category | 1237 | 97.17% |
| | | PEN [e] resistance category | 1123 | 88.22% |
| | | TAX [f] resistance category | 1203 | 94.50% |
| non-β-lactam resistance category | 3117 | CHL [g] resistance category | 3060 | 98.17% |
| | | CLI [h] resistance category | 3.074 | 98.62% |
| | | COT [i] resistance category | 2899 | 93.01% |
| | | ERY [j] resistance category | 3020 | 96.89% |
| | | LFX [k] resistance category | 3106 | 99.65% |
| | | VAN [l] resistance category | 3117 | 100.00% |
| | | RIF [m] resistance category | 2406 | 77.19% |
| | | TET [n] resistance category | 2025 | 64.97% |
| | 2431 | RIF resistance category (without intermediate calls) | 2406 | 98.97% |
| | 2082 | TET resistance category (without intermediate calls) | 2025 | 97.26% |

[a] Amoxicillin; [b] Ceftriaxone; [c] Cefuroxime; [d] Meropenem; [e] Penicillin; [f] Cefotaxime; [g] Chloramphenicol; [h] Clindamycin; [i] Co-trimoxazole; [j] Erythromycin; [k] Levofloxacin; [l] Vancomycin; [m] Rifampin; [n] Tetracycline.

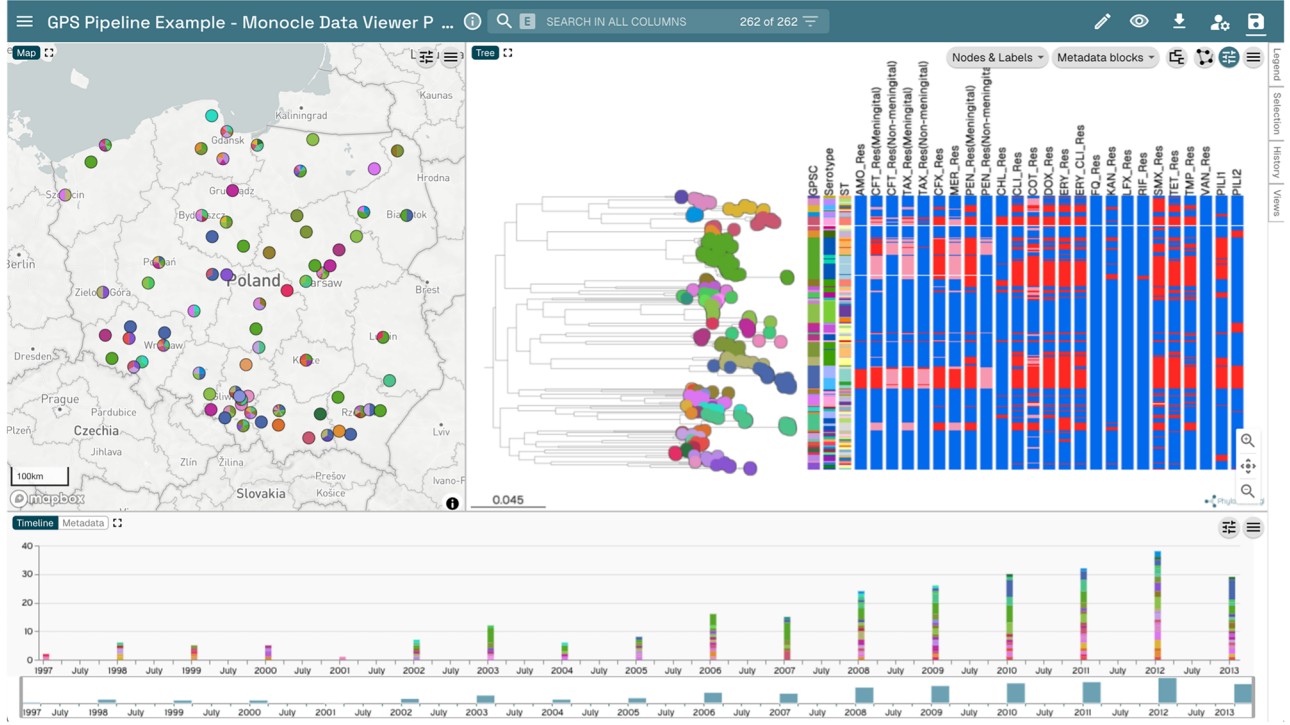

**Fig. 3 | An example visualisation of GPS Pipeline outputs combined with phylogenetic tree.** Combining with sample metadata and phylogenetic tree, the GPS Pipeline provides typing and prediction on lineage, serotype, and a wide range of antimicrobial resistance with spatial, temporal and phylogenetic context on 262 public Poland genomes. They are key public health information that have the potential to guide the selection of suitable vaccines and antimicrobials.

(through WSL), and macOS. It is also compatible with LSF-based HPC clusters and the Seqera Platform (previously known as Nextflow Tower). We tested the GPS Pipeline on Linux (Ubuntu 22.04), Windows (Windows 11 with Ubuntu 22.04 through WSL2), and macOS (Sonoma 14.4) machines with ≥16 GB RAM, as well as LSF-based HPC clusters (Wellcome Sanger Institute Farm5 and Farm22 HPC clusters), and generated consistent results. After downloading the pipeline and running its initialisation function, which downloads all required additional files and container images, the pipeline can be used without an internet connection.

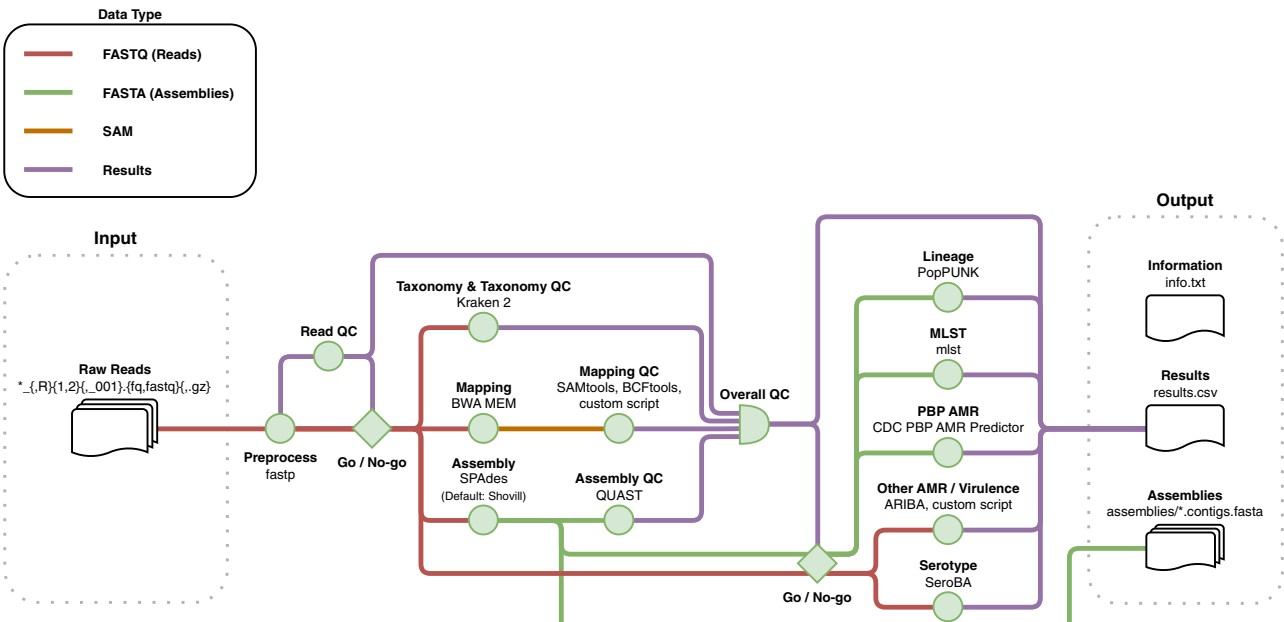

**Fig. 4 | The schematic workflow of the GPS Pipeline.** Once the user starts the GPS Pipeline with a directory path provided, the pipeline will capture all samples which raw read files match the glob pattern in that specific directory. All captured samples are processed by several bioinformatics tools and have their quality assessed by the quality control (QC) processes, which jointly determine their overall quality based on the metrics of read, taxonomy, mapping and assembly. For samples that have passed the quality control, they are subjected to further analysis by the in silico typing processes, which assign Global Pneumococcal Sequence Cluster (GPSC) lineage, and predict serotype, multilocus sequence typing (MLST) and a wide range of antimicrobial resistance (AMR) and virulence targets. The results for all samples are collated into a single comma-separated values (CSV) file. The pipeline saves the runtime information and version of the software for reproducibility, and the generated assembly of each sample.

The pipeline logic of the GPS Pipeline is programmed in Nextflow Domain-Specific Language 2 (DSL2) and supplemented by custom Apache Groovy functions. The pipeline logic invokes bioinformatics tools for analyses, and executes Shell or Python 3 scripts for data and metadata manipulations.

In this pipeline, we selected bioinformatics tools that are automated, reliable, fast, memory-efficient, and output results in a standardised text-based format that can be parsed programmatically. When possible, we selected existing bioinformatics tools that met these criteria and built custom scripts when necessary.

## Quality control

The pipeline starts with a series of quality control checks on the FASTQ files and genomes based on the configurable criteria listed in Supplementary Table 1. The pipeline first checks for corrupted FASTQ files (files that cannot be decompressed or contain incomplete data). Samples with any corrupted files are not further processed. An error message to indicate which file is corrupted is shown in the result file (results.csv).

Intact paired-end FASTQ files are then assessed by fastp v0.23.4[43]. fastp automatically performs adaptor trimming, quality filtering, and other operations to clean up the FASTQ files for downstream processing, as well as acquiring the total base count of the read files. To pass the QC, the total base count should exceed the multiplication of the minimum sequence depth and lower assembly length limit, which is 38 Mbps. Pre-processed samples that pass QC are then processed in parallel through taxonomy, mapping and de novo assembly processes as stated below for further quality control based on these metrics.

To detect any potential contamination, the pipeline runs Kraken 2 v2.1.2[44] on the FASTQ file pairs against the Minikraken v1 database[45] as the default for taxonomy classification. Kraken 2 runs faster and requires less memory when compared to Kraken 1, and it uses a capped-size database available[46], enabling it to run on the usually limited system resources available on personal computers (PCs). In

contrast, alternatives like GTDB-Tk with its Genome Taxonomy Database (GTDB)[47], have memory requirement exceeding most PCs capability (Supplementary Table 2). A QC-passed genome should have ≥60% reads mapped to *S. pneumoniae* while ≤2% reads mapped to non-pneumococci species.

To ensure the sequencing completeness and to detect a mixture of two or more pneumococcal isolates in a single sample, the FASTQ file pairs are mapped to the reference genome *S. pneumoniae* ATCC 700669 (NCBI accession no. FM211187) by default, the standard reference since the beginning of the GPS Project, using the BWA-MEM algorithm of BWA v0.7.17[48]. Its successor, BWA-MEM2, was not used due to its much higher memory requirement, rendering it unsuitable to run on PCs[49]. The output SAM files are converted into BAM files and sorted using SAMTools v1.16[50], and the reference coverage percentage is then calculated. The sorted BAM files are used to call the SNP sites, which are saved into VCF files using BCFTools v1.16[50]. Het-SNP sites are then counted by a custom Python script (bin/het_snp_count.py). This script filters out het-SNPs that are within 50-bp proximity to avoid overestimating the het-SNPs. In this step, a QC-passed genome is expected to have a coverage of the reference genome ≥60% and a count of het-SNP ≤220. The former threshold ensures sequence reads of the sample have sufficient coverage over the genome, while avoiding being over-aggressive to account for the highly diverse genome of the pneumococcus. The later threshold confirms isolates having an acceptable purity, which were established based on analysed data across 29,913 pneumococcal genomes in the GPS project (Supplementary Fig. 2).

To assess the assembled genome quality based on sequencing depth, length and number of contigs, de novo assembly is carried out using Shovill v1.1.0 by default on the FASTQ file pairs. The quality metrics are summarised by QUAST v5.0.2[51]. QC-passed genomes are expected to have a sequence depth of ≥20x, a length of assembly between 1.9−2.3 Mbps, and a number of contigs ≤500.

At the end of each QC check above, a Shell script (bin/validate_file.sh for FASTQ files integrity, bin/get_read_qc.sh for basic read quality, bin/get_taxonomy_qc.sh for taxonomy-based purity check, bin/get_mapping_qc.sh for mapping-based coverage and purity check, bin/get_assembly_qc.sh for assembly quality) assigns QC pass/fail category based on the QC metrics. Another Shell script (bin/get_overall_qc.sh), then assigns genomes that have passed all QC checks to the overall QC pass category. QC-failed genomes do not proceed downstream for in silico typing.

### De novo assembly

Shovill v1.1.0, a de novo assembler with automated optimisations, is set as the default assembler, with the included option of Unicycler v0.5.0[52] to be used by the pipeline as an alternative assembler, in case Shovill fails to generate optimal assemblies for some samples. Both assemblers are based on SPAdes. A third, Velvet-based assembler VelvetOptimiser v2.2.6, was also evaluated, but not included. The decision was based on results from tested reads from different Illumina short-read sequencing platforms (Table 3), carried out using the default parameters of the assemblers, except instructing them to use all available threads on the machine and discarding contigs with less than 500 bps. Shovill was selected as the most appropriate assembler for this pipeline based on it having the shortest runtime, lowest maximum memory usage, and ability to generate high-quality assemblies. Shovill only takes 25.3% of runtime and 70.5% of maximum memory to generate assemblies of similar quality, as compared to Unicycler, based on the number of contigs and N50 metrics, which is more than sufficient for the downstream tools in the pipeline. At the same time, VelvetOptimiser requires 172.7% of runtime compared to Shovill while generating the worst quality assemblies and is a less stable software, making it unsuitable for the pipeline.

### In silico typing

**Genomic lineage.** We assign Global Pneumococcal Sequence Clusters (GPSCs) to define lineages, which are based on genome-wide variations as compared to the conventional multilocus sequence typing (MLST) scheme, which only takes the sequences of 7 housekeeping genes into account[14,53]. GPSCs assignment is coherent by nature, allowing comparison of results between different studies. To assign GPSCs, PopPUNK v2.6.3 and the latest PopPUNK GPS database v9 (gps-project.cog.sanger.ac.uk/GPS_v9.tar.gz & gps-project.cog.sanger.ac.uk/GPS_v9_external_clusters.csv) are used.

**In silico serotyping.** For in silico serotype prediction, SeroBA v2.0.4 was selected. It has a low memory and computational power requirement, yet high prediction accuracy and specificity. These advantages are evident when compared to PneumoCat which is another common in silico pneumococcal serotyping tool[54]. The current pipeline utilises SeroBA database v2.0.4 as it could identify 102 of 107 known pneumococcal serotypes and 4 groups of non-encapsulated pneumococci. As new serotypes are detected, the pipeline can be easily updated to accommodate the latest version of SeroBA and its database.

**MLST.** mlst v2.23.0 is used to infer MLST profile and sequence type (ST) from assemblies. The current pipeline uses pneumococcal PubMLST data as of 1st July 2024. As new ST are detected, the pipeline can also be updated to include the latest version of the PubMLST database.

**Antimicrobial resistance.** The GPS Pipeline predicts antimicrobial susceptibility to 19 commonly used antibiotics using the CDC PBP AMR Predictor and ARIBA v2.14.6, and interprets the MIC predictions based on the 2014 CLSI guideline[31]. The known resistance genes and mutations are summarised in Table 5.

The CDC PBP AMR Predictor assigns PBP transpeptidase domain protein sequence types to the PBP1a, PBP2b, and PBP2x proteins encoded by draft genome assemblies, collectively referred to as PBP type. It then predicts the MICs against 6 β-lactam antibiotics, including: AMO, CFT, TAX, CFX, MER, and PEN. The machine learning model used for the prediction[7] was previously shown to yield high percent essential agreement (MICs agree within ±1 dilution) > 97% and percent category agreement (interpretive results agree) > 93% in a pneumococcal genome dataset that was not used for building the machine learning model[55]. The resistance category against each antibiotic is also inferred from the predicted MIC according to the CLSI guidelines.

ARIBA v2.14.6 is used with a pneumococcus-specific database, which we compiled to perform the AMR predictions against 13 non-β-lactam antibiotics by detecting the presence or absence of known resistance genes, mobile elements or mutations listed in Table 5. It was chosen because newly recognised resistance genes or mutations can be easily added to the ARIBA database for detection, enabling the pipeline to quickly adapt to newly discovered AMR mechanisms in the future. It should be noted that the nonsense mutations across the nucleotide range 166-201 (amino acid residue 56–67) of folP, which result in resistance to SMX[56,57], cannot be readily represented in an ARIBA database. Therefore, ARIBA is only used to detect and extract any mutations in the folP gene, and a subsequently run Python script (bin/parse_other_resistance.py) detects any disruptive mutation within the nucleotide range.

To avoid spurious detection, the presence of a resistance gene is confirmed by the same Python script (bin/parse_other_resistance.py), which determines if the gene meets the criteria of ≥ 80% coverage and ≥ 20x read depth. For a resistance mutation, a ≥ 10x sequence depth threshold is used for confirmation by this script. The categorical prediction of antimicrobial susceptibility (susceptible, intermediate, resistant) against CHL, CLI, COT, DOX, ERY, ERY and CLI, FQ, KAN, LFX, RIF, SMX, TET, TMP, and VAN is then saved to a report. The MIC range based on the categorical prediction is also added, based on the 2014 CLSI guidelines.

**Virulence factor.** The GPS Pipeline also detects the presence and absence of virulence factors, including pilus genes of PILI-1, and PILI-2, using ARIBA v2.14.6 with the identical criteria as for detecting resistance genes (≥ 80% coverage and ≥ 20x read depth).

### Containerisation

To implement the above functions in containerised environments, we included publicly available Docker images (Supplementary Table 3) developed by a communal effort led by the State Public Health Bioinformatics Group (StaPH-B)[58] whenever possible. These images are well maintained and tested to be stable, appropriately versioned (which enables version pinning and simple upgrade), and is open source. During the development of the GPS Pipeline, we have also made contributions toward this effort by updating and adding testing stages to the Dockerfiles for the Docker images of PopPUNK and ARIBA, which were promptly reviewed and added to become part of their releases. In addition, we built Docker images of CDC PBP AMR Predictor and SeroBA, as up-to-date images were not available. To handle the extensive usage of command line utilities and Python scripts in the pipeline, the Network-Multi Tool Docker image by WBITT[59] and the Pandas Docker image by Alexander Mancevice[60] were chosen, respectively.

### Reporting summary

Further information on research design is available in the Nature Portfolio Reporting Summary linked to this article.

**Table 5 | Tools and mechanisms deployed by the pipeline to predict antimicrobial resistances and virulence markers**

| Tool | Type | AMR / Virulence Marker | Mechanism |
|---|---|---|---|
| CDC PBP AMR Predictor | Antimicrobial Resistance | Amoxicillin (AMO) | Based on the amino acid sequence signatures within transpeptidase domains of three penicillin-binding proteins (PBP1a, PBP2b, PBP2x) via a machine learning model[55,62] |
|  |  | Ceftriaxone (CFT) |  |
|  |  | Cefotaxime (TAX) |  |
|  |  | Cefuroxime (CFX) |  |
|  |  | Meropenem (MER) |  |
|  |  | Penicillin (PEN) |  |
| ARIBA + Custom script* | Antimicrobial Resistance | Chloramphenicol (CHL) | Presence of *cat, catpC194, catpC233,* or *catQ* gene[63,64] |
|  |  | Erythromycin and Clindamycin (ERY and CLI) ^ | A2061G substitution in 23S rRNA, or presence of *ermB, ermC, ermTR* or *ermBL* gene[7,9,65] |
|  |  | CLI | Inferred from ERY and CLI above |
|  |  | ERY | In addition to inferred from ERY and CLI above, the presence of the *mef(A), mef(E),* or *msr(D)* gene[66] |
|  |  | Fluoroquinolones (FQ) | S81C/F/I/Y, E85K or Q118A substitution in *gyrA* gene, E474K substitution in *gyrB* gene, A63T, S79F/L/Y or D83G/N substitution in *parC* gene, or D435H/N, P454S or E474K substitution in *parE* gene[67,68] |
|  |  | Levofloxacin (LFX) | Inferred from FQ above |
|  |  | Kanamycin (KAN) | Presence of *aph3'-III* gene[69] |
|  |  | Rifampin (RIF) | D489E/N or H499N substitution in *rpoB* gene[70] |
|  |  | Sulfamethoxazole (SMX) | Any disruptive mutation within nucleotide range 166-201 of the *folP* gene[56,57] |
|  |  | Trimethoprim (TMP) | I100L substitution in *folA* gene[7] |
|  |  | Co-Trimoxazole (COT) | Inferred on SMX and TMP above: resistance to either SMX or TMP is reported as intermediate, and resistance to both is reported as resistance |
|  |  | Tetracycline (TET) | Presence of *otr(A), tet(M), tet(S), tet(S/M),* or a selected group of *tet* genes[71] |
|  |  | Doxycycline (DOX) | Inferred from TET above |
|  |  | Vancomycin (VAN) | Presence of *vanA, vanB, vanC, vanD, vanE,* or *vanG* gene[72] |
|  | Virulence Marker | PILI-1 | Presence of *rrgA* gene[73] |
|  |  | PILI-2 | Presence of *pitB* gene[74] |

*A custom Python script (github.com/GlobalPneumoSeq/gps-pipeline/blob/master/bin/parse_other_resistance.py) was written to supplement the functionality of ARIBA. ARIBA can only report the detection of gene presence, absence or mutations, and is unable to efficiently detect disruptive mutations across a range, therefore, the Python script interprets the detection results into antimicrobial resistance (AMR) predictions and has a dedicated logic for folP mutations within a target range to predict AMR against SMX.
^"ERY and CLI" is not an antibiotic, but serve as a reference to indicate whether a sample is predicted to be resistant to both ERY and CLI.

## Data availability

Published data from the GPS Database is available on Monocle Data Viewer at data.monocle.sanger.ac.uk and associated sequence read files are searchable and downloadable in the European Nucleotide Archive at ebi.ac.uk/ena via their ERR accession numbers. The list of accession numbers is available in Supplementary Data 3. This study did not publish new data, and all data analysed were previously published.

## Code availability

The GPS Pipeline is available on GitHub at github.com/GlobalPneumoSeq/gps-pipeline, the version of the code used in this study is registered on Zenodo[61].

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

## Acknowledgements

This work is funded by the Bill & Melinda Gates Foundation (grant code INV-003570; awarded to S.D.B.) and Wellcome Trust (grant reference 206194; awarded to S.D.B.) through the Global Pneumococcal Sequencing project.

## Author contributions

H.C.H.H. developed and documented the GPS Pipeline, carried out the validation, and wrote the manuscript. H.C.H.H., S.D.B. and S.W.L. conceptualised the project. N.K. collated the ARIBA AMR database and contributed to the development of the mapping QC module and non-β-lactam AMR module. V.D. contributed to the development of the lineage module. C.Y., B.M., Y.L., P.A.H. and L.M. contributed to the development of the β-lactam AMR module. S.D.B. and S.W.L. supervised the project. All authors contributed to the manuscript.

## Competing interests

The authors declare no competing interests.
