## [Peer Review file · Nature Communications]

GPS Pipeline: portable, scalable genomic pipeline for *Streptococcus pneumoniae* surveillance from Global Pneumococcal Sequencing Project

Corresponding Author: Dr Harry C. H. Hung

Version 0:

Reviewer comments:

Reviewer #1

(Remarks to the Author)

The manuscript introduces an innovative tool, the GPS Pipeline, designed to enhance the genomic surveillance of *Streptococcus pneumoniae*. Utilizing Docker and Singularity, this tool aims to simplify genomic analysis, making it accessible to a broader audience, including those in resource-limited settings. While the initiative is commendable, the manuscript could significantly benefit from addressing compatibility with various sequencing technologies more thoroughly and providing a deeper comparison with existing genomic tools.

1) The demonstration of the GPS Pipeline's utility in processing extensive genomic datasets effectively highlights its potential to impact public health surveillance positively. The strategic use of containerization technologies is particularly notable for its role in facilitating easier access to genomic analysis tools. To further enhance the manuscript, it would be beneficial to include detailed examples showcasing the tool's output. These examples could include case studies or visual outputs from the data analysis, which would help readers visualize how the tool functions in real-world scenarios and the kind of results it produces.

2) The tool is described as supporting only Illumina paired-end short reads. This specification may significantly limit the tool's applicability given the diversity of sequencing technologies widely used in the field of genomics. The discussion should extend to address whether the pipeline could potentially be compatible with fastq files from other sequencing platforms, including short-read technologies like AVITI by Element Biosciences, MGI-seq, and Ion Torrent, as well as long-read platforms such as Oxford Nanopore and PacBio. The authors should explore and discuss the technical feasibility and any necessary modifications required to process data from these technologies. Additionally, it is crucial for the manuscript to explicitly state these limitations and discuss any plans to adapt the tool to handle these diverse data types.

3) While the manuscript outlines the need for advanced tools in genomic surveillance, it lacks a comprehensive discussion of several recent developments in the field, such as PneusPage. An updated literature review including these advancements could provide a broader context for the GPS Pipeline and help justify its development. Comparing the features, usability, and access of recent tools like Pathogenwatch and PneusPage alongside the GPS Pipeline would enrich the manuscript, highlighting the unique contributions and positioning of the GPS Pipeline within the current technological landscape.

4) There is a lack of comparative analysis with other tools such as the SPN NextFlow Pipeline, Bactopia, and Pathogenwatch. Including such an analysis would be crucial for objectively assessing the GPS Pipeline's performance relative to these established benchmarks. The manuscript should detail a comparison focusing on various aspects such as accuracy, ease of use, computational efficiency, and the adaptability of each tool to different types of sequencing data.

(Remarks on code availability)

Reviewer #2

(Remarks to the Author)

In this manuscript, Hung et al. describe a bioinformatic pipeline for the analysis of sequence data from isolates of *Streptococcus pneumoniae*. The pipeline, starting from raw reads, performs quality control, assembly, serotype prediction, antibiotic resistance prediction and molecular typing using schema commonly used in pneumococcal epidemiology. The pipeline is written using nextflow, a widely used software platform for workflow assembly, and uses containers in order to make it portable across computer architectures and operating systems. The system is validated using a large dataset from the Global Pneumococcal Sequencing (GPS) project. The authors claim that the pipeline has been specifically optimized and tailored to be used in areas lacking stable internet connection and computational facilities.

The tool reported here is certainly useful for epidemiological studies in *S. pneumoniae*, however I have a few concerns. First of all, it is not clear to me what the novelty is. As far as I understand, the authors wrapped a set of standard tools in a bioinformatic pipeline using nextflow and docker. As I already mentioned, this kind of effort is extremely useful and is now commonplace in the bioinformatic community. There are nowadays large repositories of workflows written using nextflow that allow researchers with basic bioinformatic skills to run complex analyses, like, for instance, the nf-core repository (<https://nf-co.re/>). Second, I am not convinced by the way in which the performances of this pipeline are evaluated. As far as I understand, isolate's features such as serotype and antibiotic resistance predicted in silico by the pipeline are compared to data contained in the GPS database that are also predicted in silico. As such, this comparison just proves the consistency of the nextflow implementation, not the predictive power of the pipeline. It is relevant as a preliminary check, but does not tell us anything about the relevance of the results. If my understanding is correct, it is not surprising that there is a high degree of concordance between original data and results, and reporting the data in this way might give a biased picture of the overall performance of the system when compared to experimental results.

(Remarks on code availability)

Version 1:

Reviewer comments:

Reviewer #1

(Remarks to the Author)

Thank you for your thorough revision. The manuscript has improved significantly, and the major concerns raised in the initial review have been addressed satisfactorily.

The addition of a real-world example using Microreact and the description of broader adoption strengthen the demonstration of the pipeline's utility. The inclusion of an updated introduction, incorporating recent tools such as PneusPage, and the addition of a comparative table provide a clearer context for the GPS Pipeline within the current bioinformatics landscape. The explicit discussion of the Illumina-only limitation, with justification and future plans, improves transparency and sets realistic expectations for potential users.

While compatibility with other short-read platforms (such as AVITI, MGI-seq, Ion Torrent) is not fully explored, the discussion now clearly acknowledges this limitation and provides a reasonable rationale based on current usage patterns. Given that Illumina remains dominant for pneumococcal sequencing, this limitation does not reduce the manuscript's relevance or utility.

Overall, the revisions have substantially improved clarity, context, and completeness. I have no further major concerns and support publication in its current form.

(Remarks on code availability)

Reviewer #2

(Remarks to the Author)

I appreciate that the authors included a comparison between the AMR predicted results from their pipeline and experimental data. I have no more comments

(Remarks on code availability)

Reviewer #1

Reviewer's original comment

The manuscript introduces an innovative tool, the GPS Pipeline, designed to enhance the genomic surveillance of *Streptococcus pneumoniae*. Utilizing Docker and Singularity, this tool aims to simplify genomic analysis, making it accessible to a broader audience, including those in resource-limited settings. While the initiative is commendable, the manuscript could significantly benefit from addressing compatibility with various sequencing technologies more thoroughly and providing a deeper comparison with existing genomic tools.

Author's response

We appreciate the reviewer's recognition of our work as a commendable initiative. We address the concerns raised below in a point-by-point manner.

Reviewer's original comment

1) The demonstration of the GPS Pipeline's utility in processing extensive genomic datasets effectively highlights its potential to impact public health surveillance positively. The strategic use of containerization technologies is particularly notable for its role in facilitating easier access to genomic analysis tools. To further enhance the manuscript, it would be beneficial to include detailed examples showcasing the tool's output. These examples could include case studies or visual outputs from the data analysis, which would help readers visualize how the tool functions in real-world scenarios and the kind of results it produces.

Author's response

Thank you for the insightful suggestion. We have now created a Microreact showcase instance based on publicly available pneumococcal genomes from Poland (via the Monocle Data Viewer), using the GPS Pipeline outputs, sample metadata, and additional phylogenetic analysis. This example illustrates how the pipeline provides extensive contextual information for genomic surveillance. Details of this showcase are included in a new subsection in the "Results" section.

Additionally, we have expanded the "Results" section with a new subsection that describes real-world applications of the GPS Pipeline, including its role in the GPS project and its adoption by other researchers, further demonstrating its utility in public health surveillance.

Reviewer's original comment

2) The tool is described as supporting only Illumina paired-end short reads. This specification may significantly limit the tool's applicability given the diversity of sequencing technologies widely used in the field of genomics. The discussion should extend to address whether the pipeline could potentially be compatible with fastq

files from other sequencing platforms, including short-read technologies like AVITI by Element Biosciences, MGI-seq, and Ion Torrent, as well as long-read platforms such as Oxford Nanopore and PacBio. The authors should explore and discuss the technical feasibility and any necessary modifications required to process data from these technologies. Additionally, it is crucial for the manuscript to explicitly state these limitations and discuss any plans to adapt the tool to handle these diverse data types.

Author's response

We agree that limiting support to Illumina paired-end short reads reduces the pipeline's versatility. However, according to ENA records, the vast majority (99.6%, n=42,291/42,452) of publicly available pneumococcal genomes in 2024 were still generated using Illumina platforms. As such, the current limitation has a relatively minor practical impact.

We have added a new paragraph to the "Discussion" section to explicitly state this limitation, explain the rationale for this design choice, quantify its current impact, describe potential workarounds, and outline future plans for broader compatibility.

Reviewer's original comment

3) While the manuscript outlines the need for advanced tools in genomic surveillance, it lacks a comprehensive discussion of several recent developments in the field, such as PneusPage. An updated literature review including these advancements could provide a broader context for the GPS Pipeline and help justify its development. Comparing the features, usability, and access of recent tools like Pathogenwatch and PneusPage alongside the GPS Pipeline would enrich the manuscript, highlighting the unique contributions and positioning of the GPS Pipeline within the current technological landscape.

Author's response

PneusPage is a recent development that was not available at the time of our original submission. We agree that including it enhances the manuscript's relevance. Accordingly, we have added a paragraph in the "Introduction" section discussing PneusPage's features and limitations.

We had already discussed Pathogenwatch in the same section, and the additional context now provides a clearer landscape of existing tools. A more detailed comparison is provided in response to the next point.

Reviewer's original comment

4) There is a lack of comparative analysis with other tools such as the SPN NextFlow Pipeline, Bactopia, and Pathogenwatch. Including such an analysis would be crucial for objectively assessing the GPS Pipeline's performance relative to these

established benchmarks. The manuscript should detail a comparison focusing on various aspects such as accuracy, ease of use, computational efficiency, and the adaptability of each tool to different types of sequencing data.

Author's response

We have added a new table to provide comprehensive comparison between the GPS Pipeline and other established tools, which covers key features. We hope this addition provides the reviewer and readers with a clearer understanding of the GPS Pipeline's strengths and its unique position within the current landscape.

Reviewer #2

Reviewer's original comment

In this manuscript, Hung et al. describe a bioinformatic pipeline for the analysis of sequence data from isolates of *Streptococcus pneumoniae*. The pipeline, starting from raw reads, performs quality control, assembly, serotype prediction, antibiotic resistance prediction and molecular typing using schema commonly used in pneumococcal epidemiology. The pipeline is written using nextflow, a widely used software platform for workflow assembly, and uses containers in order to make it portable across computer architectures and operating systems. The system is validated using a large dataset from the Global Pneumococcal Sequencing (GPS) project. The authors claim that the pipeline has been specifically optimized and tailored to be used in areas lacking stable internet connection and computational facilities.

Author's response

We thank the reviewer for their careful summary of the manuscript. We appreciate the recognition of the GPS Pipeline's structure and intended context of use. We address the reviewer's specific concerns below in a point-by-point manner.

Reviewer's original comment

The tool reported here is certainly useful for epidemiological studies in *S. pneumoniae*, however I have a few concerns. First of all, it is not clear to me what the novelty is. As far as I understand, the authors wrapped a set of standard tools in a bioinformatic pipeline using nextflow and docker. As I already mentioned, this kind of effort is extremely useful and is now commonplace in the bioinformatic community. There are nowadays large repositories of workflows written using nextflow that allow researchers with basic bioinformatic skills to run complex analyses, like, for instance, the nf-core repository ([https://nf-co.re/\[nf-co.re\]](https://nf-co.re/[nf-co.re])).

Author's response

While the GPS Pipeline does use Nextflow, Docker, and standard bioinformatics tools, its novelty lies in how these are integrated into a ready-to-run solution tailored for pneumococcal analysis. It is designed to work out-of-the-box for researchers with only basic computer skills, without needing external infrastructure. We have slightly revised the last paragraph of the “Discussion” section to highlight this.

Another key novelty is that we have not only developed the pipeline but also deployed it across multiple partners in Africa, Asia, and Latin America, where teams can successfully run analyses on their locally generated data. This breakthrough enables sustainable, decentralised pneumococcal surveillance by empowering regional laboratories to conduct genomic analysis without relying on external infrastructure, thereby strengthening long-term monitoring and rapid response capacity. We therefore believe the pipeline offers substantial practical impact alongside its novel technical contribution.

We chose not to include nf-core workflows in our tool comparison, as none currently provide pneumococcal-specific analyses.

Reviewer's original comment

Second, I am not convinced by the way in which the performances of this pipeline are evaluated. As far as I understand, isolate's features such as serotype and antibiotic resistance predicted *in silico* by the pipeline are compared to data contained in the GPS database that are also predicted *in silico*. As such, this comparison just proves the consistency of the nextflow implementation, not the predictive power of the pipeline. It is relevant as a preliminary check, but does not tell us anything about the relevance of the results. If my understanding is correct, it is not surprising that there is a high degree of concordance between original data and results, and reporting the data in this way might give a biased picture of the overall performance of the system when compared to experimental results.

Author's response

We understand the concern that comparing GPS Pipeline outputs to the *in silico* results in the GPS database could appear to involve circular reasoning. To address this concern, we have added a new subsection and a table into the “Results” section, directly comparing GPS Pipeline *in silico* predictions to experimental phenotype data available in the GPS database to demonstrate the high concordance.

Moreover, as mentioned in the “Validation” subsection of the manuscript, the *in silico* results in the GPS database have previously been validated against experimental phenotypic data in a published study, demonstrating high concordance. Additionally, the GPS Pipeline employs modern and more efficient tools than those originally used to generate the original GPS database results.

Therefore, this comparison is not simply a check of Nextflow re-implementation consistency, but a way to show that the pipeline continues to produce results concordant with trusted reference data, despite changes in the underlying tools.